# A Zest of LIME: Towards Architecture-Independent Model Distances

**Hengrui Jia, Hongyu Chen, Jonas Guan**
University of Toronto and Vector Institute
{nickhengrui.jia, hy.chen}@mail.utoronto.ca, jonas@cs.toronto.edu

**Ali Shahin Shamsabadi**
Vector Institute and The Alan Turing Institute
a.shahinshamsabadi@turing.ac.uk

**Nicolas Papernot**
University of Toronto and Vector Institute
nicolas.papernot@utoronto.ca

## Abstract

Definitions of the distance between two machine learning models either characterize the similarity of the models' predictions or of their weights. While similarity of weights is attractive because it implies similarity of predictions in the limit, it suffers from being inapplicable to comparing models with different architectures. On the other hand, the similarity of predictions is broadly applicable but depends heavily on the choice of model inputs during comparison. In this paper, we instead propose to compute distance between black-box models by comparing their Local Interpretable Model-Agnostic Explanations (LIME). To compare two models, we take a reference dataset, and locally approximate the models on each reference point with linear models trained by LIME. We then compute the Cosine distance between the concatenated weights of the linear models. This yields an approach that is both architecture-independent and possesses the benefits of comparing models in weight space. We empirically show that our method, which we call Zest, helps in several tasks that require measurements of model similarity: verifying machine unlearning, and detecting many forms of model reuse, such as model stealing, knowledge distillation, and transfer learning.[1]

## 1 Introduction

We explore the problem of quantifying similarity between machine learning (ML) models with distance metrics, with a focus on deep neural networks (DNNs). Comparing the functional behavior of ML models is often challenging because they are not easily inspectable (e.g. located in the cloud). In addition, ML models that capture similar knowledge may not share similar architectures, and vice versa, making it difficult to compare models in weight space. Finally, ML models that solve different tasks may nonetheless share similar knowledge, such as in the case of transfer learning (Li et al., 2021). This is emphasized in DNNs, which are not only complex, often containing millions of parameters, but also difficult to interpret (Ribeiro et al., 2016).

Previous methods for measuring model distances use either the similarity of predictions (Li et al., 2021), or the similarity of weights (Jia et al., 2021). However, methods that measure the similarity of predictions only capture the local behavior of models at each data point, and it is hard to choose a set of inputs to compare their global behaviors. This causes high variance in the results: models that are unrelated may happen to give very similar predictions on some inputs, while related models (e.g., fine-tuned from one another) may give different predictions on other inputs. Methods that measure the similarity of weights are more complete, because they capture the global behavior of models, and weight similarity implies prediction similarity in the limit: as the weights of the compared models approach each other, so will their predictions. But, to be meaningful, weight-space methods can only be applied when the compared models share the same architecture. Furthermore, they require white-box access to the models to compare weights. These constraints restrict their use in practice.

---

[1]Code is available at: https://github.com/cleverhans-lab/Zest-Model-Distance

Instead, we propose to compute distances between models by first approximating their global behavior through a collection of linear models. We name our approach Zest. We start by applying the Local Interpretable Model-Agnostic Explanations (LIME) algorithm of Ribeiro et al. (2016) to generate a collection of linear models that each approximates some local behavior of a compared model. We chose LIME because it is black-box (only requires access to the models' predictions) and architecture-independent, and as a consequence so is Zest. The collection of these local approximations form a global approximation of model behavior. Since the approximation models obtained are linear, they are straightforward to compare. Indeed, to measure similarities and differences between two models being compared, we compute the Cosine distance between the concatenated weights of their respective collections of approximated linear models. Because we compare local linear approximations of the models rather than their predictions directly, this allows Zest to interpolate between the benefits of prior approaches in the weight or prediction spaces described earlier.

In particular, our empirical validation of Zest demonstrates that our approach is not sensitive to the choice of points used by LIME to locally approximate the models being compared. This departs from prior approaches that compared model predictions directly and as a result were overly sensitive to the choice of points the comparison is made on. This characteristic allows Zest to correctly capture the differences and similarities between pairs of unrelated or related classifiers.

Furthermore, we show that the distance computed by Zest finds natural applications in detecting model reuse (*e.g.,* model stealing, knowledge distillation, and transfer learning) and verifying machine unlearning. Li et al. (2021) first proposed to introduce a distance to inform the detection of classifiers which were trained by reusing another classifier. In particular, this is the case in model extraction where an adversary observes the predictions of a model to reproduce its behavior in a stolen copy of the model. Whereas Li et al. (2021) left the identification of stolen copies of a model as future work after failing to apply their proposed ModelDiff metric to this problem, we find it can be solved by Zest. In a completely different setting, we also find an application of Zest to machine unlearning: the process of removing the effects of learning data points from a model (Cao & Yang, 2015). We use Zest as a heuristic to inform whether or not a user's data has been unlearned, by comparing the decision boundaries of the models before and after unlearning around that data.

Our main contributions are as follows:

- We propose Zest, an approach for computing an architecture-independent distance metric that measures the similarity between ML models. Zest only requires black-box access, and compares the global behavior of the models, which it approximates using LIME.

- We validate that Zest measures model similarity by showing that the distance between different epochs of the same model is on average $99.27(\pm0.19)\%$ on CIFAR-10, $98.75(\pm0.12)\%$ on CIFAR-100, $92.68(\pm2.22)\%$ on AG News and $81.17(\pm3.01)\%$ on Speech Commands, closer than the distance between two models with different initialization trained on the same dataset. We run our experiments in the vision, text and audio domains using the ResNet18, ResNet20, ResNet50, LSTM, M5, and MobileNet-v2 architectures, on the CIFAR-10, CIFAR-100, AG News, Speech Commands, ImageNet, Flower102 and SDog120 datasets.

- We show that Zest can be used to help detect model reuse, including the hard case of model stealing, and inform unlearning verification. The distance between a model stolen via model extraction and the original model is $19.96(\pm4.28)\%$ on CIFAR-10 and $54.87(\pm1.70)\%$ on CIFAR-100, closer on average than the distance between the original and a model retrained on the same datasets. Compared to ModelDiff, which had a 0% accuracy at detecting model extraction, we have 100% accuracy for the attacks in the ModelReuse benchmark (Li et al., 2021). We further show that Zest helps identify all other methods of model reuse in the benchmark. To inform unlearning verification, we show that the distance from a model to its retrained counterpart without the unlearned data is on average $144.56(\pm15.39)\%$ on CIFAR-10 and $140.51(\pm9.10)\%$ on CIFAR-100 further than the distance to a retrained model that did not exclude the data to unlearn.

## 2 BACKGROUND AND PROBLEM STATEMENT: DEFINING MODEL DISTANCE

We aim to compute a distance between two ML models $C_1$ and $C_2$ to characterize their similarity, given only the ability to observe the models' predictions on chosen inputs. We make no assumptions on the models' architectures, except that they have the same input dimensions. In other words, we only assume that they model similar distributions (e.g. traffic signs), but the compared models can

be trained for different tasks (e.g. classifying German and American traffic signs respectively). We denote the output space of the two models $D_1$ and $D_2$. We focus on DNNs, due to their popularity and the difficulty to interpret their behavior, but our distance metric applies to other ML models. For ease of exposition, we use the classification of images as a sample task throughout our discussion in §2 and §3. We also evaluate Zest on the text and audio domains and provide our results in §4.

Past work on measuring model similarity has either defined distance by comparing model weights or comparing model predictions. We briefly overview both approaches and their current limitations, then give an outline of the LIME algorithm and how it can be applied to address these limitations.

**Comparing model weights.** The weights of a parameterized model define its behavior and it can be seen how in the limit having close weights implies that two models will have close behavior; thus, one way to compare the similarity of models is to compare the difference in their weights.

One example of weight space comparison is proof of learning: Jia et al. (2021) use the comparison of model weights throughout the training procedure to provide evidence that a party has indeed trained a model. This is useful for ownership resolution. By saving checkpoints of the model during training, and showing that the model weights between subsequent checkpoints are close, the owner can provide a history that proves they are the party who trained the model.

However, comparing model weights is highly restrictive in practice: it requires both access to model weights, and for compared models to share the same architecture, so that the distance between each corresponding weight value can be compared. Even if models share the same architecture, it can be seen that two models whose weights are permutations of one another on a single layer would make exactly the same predictions yet have a non-zero distance in weight space.

**Comparing model predictions.** On the other hand, a model's behavior can also be defined by its predictions on inputs. The advantage of comparing model similarity with predictions is that we can treat the models as black-boxes. Because we only need to observe input-output pairs, we do not need access to model weights and are not restricted by the architecture of the models being compared.

In theory, if we can query a model with every input in the domain and get its predictions, we can learn its exact functional behavior, but this is intractable in practice. To address this, current approaches resort to selecting smaller sets of inputs as reference, and comparing the models based on their predictions on the reference. The disadvantage of these approaches is that (a) they can only capture point-wise behavior of the models at the chosen inputs, and (b) choosing a representative set of inputs is hard. If the chosen inputs are far from decision boundaries then even unrelated well-trained DNNs give similar predictions; whereas inputs near decision boundaries are sensitive to small changes of the boundaries, so related models (e.g. a model and its transferred copy) may predict differently.

This is consistent with findings in deep learning testing, a class of methods that aim to identify areas of the input space where supposedly similar models disagree with each other on the output space. DiffChaser (Xie et al., 2019), DeepXplore (Pei et al., 2017) and TensorFuzz (Odena et al., 2019) are three recent and popular testing frameworks. These are not applicable in our case because they focus on the differences between closely related models, whereas we are looking for a distance metric that is able to provide information on both the similarity *and* differences between any kinds of models.

Perhaps the closest to our work, ModelDiff (Li et al., 2021) is a model distance metric designed to detect model reuse, which is defined as training a model based on another to save on training costs. ModelDiff feeds a set of paired test inputs to the compared models, where each pair is selected to be close in the input space, but far in the output space. It then compares the total Cosine similarity of the distance between the outputs of each pair to find the model distance. Using this distance metric, Li et al. (2021) detected transfer learning and model compression, however, they fail at the harder case of detecting model extraction and leave it to future work. Our approach solves this open problem, and our empirical study demonstrates that our distance metric is more precise and less sensitive to changes of the model that do not affect its performance on in-distribution data.

**LIME.** Local Interpretable Model-agnostic Explanations generates interpretable linear models to explain the local behavior of ML models (Ribeiro et al., 2016). LIME learns linear models by segmenting the input space, then systematically masking segments to identify features that are most influential to the classification of a data point. For example, this implies masking contiguous areas of pixels for image classification tasks. Using the most influential features, LIME learns a linear model that approximates the decision boundary of the compared model near the data point.

Figure 1: An overview of the Zest approach to computing the distance between two black-box classifiers $C_1$ and $C_2$. Phase 1 randomly samples several reference images. Phase 2 segments the reference images into components and their location masks using a segmentation model. Phase 3 approximates the local decision boundaries around each reference image by training a LIME model that takes mask locations as input and predicts the classifiers response on these images' components. Finally, Phase 4 combines the local decision boundaries of all reference points to approximate the global decision boundaries of $C_1$ and $C_2$, and computes their Cosine distance.

**Intuition of comparing in weight space of LIME.** From the perspective of the decision boundaries, weights of a DNN capture the entire boundaries, just like how $x^2 + y^2 = 1$ captures a unit circle. This leads to high precision but restricts generality. For example, it is hard to define how $x^2 + y^2 = 1$ similar is to $max(x, y) = 1$. On the other hand, each prediction only reveals one specific mapping from the input to the output space, and multiple predictions are needed to identify a tiny region of the decision boundaries. If we draw an analogy to the Cartesian coordinate system, comparing model predictions is like comparing points in the coordinate system–the method is more general, but both a unit circle and a unit square can have points like $[0, 1]$. The method will not be precise if we compare the points and conclude the circle is similar to the square.

At a high level, we are dealing with the trade-off between precision and generality. Comparing predictions gives us general applicability whereas comparing weights gives us precision. Thus, to interpolate between these two extremes we choose to compare linear models that locally approximate the models to be compared. Using the same analogy, comparing LIME of two DNNs is similar to comparing lines that are tangent to the shape, instead of the formula of the shape (comparing weights) or points on the shape (comparing predictions). We choose the LIME algorithm for generating local approximations because it is architecture-independent, only requires black-box access, and is simple to understand. § 3 contains a more formal description of how we apply LIME.

## 3   METHOD: MEASURING DISTANCES BETWEEN MODELS WITH ZEST

Building on the insights of § 2, we now introduce an approach that is able to interpolate between the benefits of weight-space and prediction-space comparisons of two classifiers. We propose Zest, a model distance metric that measures the similarity of two models $C_1$ and $C_2$ by comparing their global behaviour as approximated by LIME (see Figure 1 for an overview).

Algorithm 1 describes our proposed model distance, which consists of four sequential phases: 1) Sampling a reference dataset; 2) Sampling nearby images as done in LIME; 3) Learning the behaviour of $C_1$ and $C_2$ locally around the reference data points with linear classifiers as done in LIME; 4) Approximating a global perspective of $C_1$ and $C_2$ by combining these linear models, and computing the distance between the combinations of linear models corresponding to $C_1$ and $C_2$. Next, we describe each phase of our proposed model distance in detail.

**Sampling representative reference data points.** We randomly sample $N$ reference images $\mathcal{X} = \{\mathbf{x}_i\}_{i=1}^N$ from the distribution task. Note that $N$ is much smaller than the size of the training set of the classifiers being compared. For example, we select $N = 128$ reference images from $50,000$ training images of CIFAR-10. In § 4.3, we show empirically that the choice of $N$ does not qualitatively change the outcome of our model distance metric computations—unlike previous approaches described in § 2 that directly compared the predictions of two classifiers. Increasing $N$ simply reduces the variance of our metric, which is intuitive because the comparison is made on more points sampled from the underlying data distribution; so reference points are more likely to cover the same local regions of the classifiers' decision surface as $N$ increases.

**Learning the local behaviours.** While the classifiers being compared can be non-linear (*e.g.,* deep neural networks in our case), the intuition behind LIME is that their behavior can be approximated *locally* by a linear model. We learn the local behaviour of each of the two classifiers being compared around each reference image by training a LIME linear regression model that captures the classifier responses to the image's components (i.e. contiguous patches of pixels denoted a super-pixel).

Similar to the original LIME paper, we create and sample image components as follows. First, we group the pixels of a reference image in $K$ super-pixels using Quickshift (Vedaldi & Soatto, 2008) and create binary super-pixel masks $\mathcal{B} = \{\mathbf{b}_k\}_{k=1}^K$, where each super-pixel binary mask $\mathbf{b}_k \in \{0,1\}^{w \times h}$ specifies the location of pixels belong to the $k$-th super-pixel. These super-pixels each represent a contiguous patch of similar pixels, like a patch of fur or grass, whose presence or absence is controlled by the binary mask. A linear model on these super-pixels provides an interpretable approximation of the local decision boundary. For example, if the classifier predicts "dog" when the patch of fur is present, but changes its prediction when it is not, then we can interpret that the patch of fur greatly influences its prediction.

Then, we create the reference image's component $\hat{\mathbf{x}} = \mathbf{x} \cdot \mathbf{m}$ by randomly choosing several super-pixels identified by 1s in component mask $\mathbf{m} \in \{0,1\}^{w \times h}$ that is combination of their corresponding binary mask. In order to explore the local area around each reference image, we repeat this process and draw $L$ different components $\hat{\mathcal{X}} = \{\hat{\mathbf{x}}_l\}_{l=1}^L$ using $L$ different component mask $\mathcal{M} = \{\mathbf{m}_l\}_{l=1}^L$. Following the original LIME paper, we set $L = 1000$.

We query classifiers to obtain their predictions on these $L$ image components as $\mathcal{Z}_1 = C_1(\hat{\mathcal{X}})$ and $\mathcal{Z}_2 = C_2(\hat{\mathcal{X}})$. Then, we train a linear regression model $I : \mathcal{M} \to \mathcal{Z}$ per classifier to have high local fidelity with respect to the classifier around the reference image $\mathbf{x}$. We call the weights of the trained regression model the local signature of its corresponding classifier.

**Approximating a global perspective.** In total, we learn $N$ linear regression models, capturing the behaviour of the classifier locally around the $N$ reference images. We approximate the global behaviour of each classifier by concatenating its $N$ local signatures $\mathcal{S}_1 = \{W_1^n\}_{n=1}^N$. Note that local signatures of each classifier have as many rows as the number of classes of the classifier. In order to compare signatures of classifiers with different number of classes when the relationship between classes output by the two classifiers is unknown, we introduce class alignment to re-position and reshape the rows of local signatures. Let *small* and *big* local signature be the local signatures of the classifiers with the lower and higher number of classes respectively. In class alignment, we start by the first row of the *small* local signature and compute its distance with each row in the *big* local signature. Then, we move the row corresponding to the minimum distance to the first row of of *big* local signature. We repeat this for all other rows in the *small* local signature while removing aligned rows of the *big* local signature from this process. Finally, we remove any remaining row of the *big* local signature not selected previously. See Appendix B for further details.

**Computing the distance.** As the global signature of classifiers are the same size, we compute the model distance as $d = Distance(\mathcal{S}_1, \mathcal{S}_2)$. In Appendix G, we show that the choice of $Distance(\cdot, \cdot)$ does not affect the performance of Zest. Our experimental results in the remainder of the paper are thus presented with the Cosine distance being used to compare global signatures.

---

**Algorithm 1:** Proposed model distance approach, Zest

---

**Input:** Black-box $D_1$-class classifier $C_1$, Black-box $D_2$-class classifier $C_2$, Reference image set $\mathcal{X} = \{\mathbf{x}_i\}_{i=1}^N$, Quickshift segmentation algorithm, Random-Selector, Linear regression training algorithm $T(\cdot, \cdot)$ and distance $Distance(\cdot, \cdot)$
**Output:** Distance, $d$, between black-box classifiers $C_1$ and $C_2$

1: $\mathcal{S}_1, \mathcal{S}_2 = \{\}, \{\}$                    ▷ Initialize model signatures
2: **for** $\mathbf{x} \in \mathcal{X}$ **do**
3:    $\mathcal{B} = \{\mathbf{b}_k\}_{k=1}^K = $ Quickshift$(\mathbf{x})$           ▷ Generate $K$ super-pixel masks
4:    $\mathcal{M} = \{\mathbf{m}_l\}_{l=1}^L = $ Random-Selector$(\mathcal{B})$     ▷ Generate $L$ image's component masks
5:    $\hat{\mathcal{X}} = \{\hat{\mathbf{x}}_l : \hat{\mathbf{x}}_l = \mathbf{x} \cdot \mathbf{m}_l\}_{l=1}^L$          ▷ Generate $L$ image's components
6:    $\mathcal{Z}_1 = C_1(\hat{\mathcal{X}}), \mathcal{Z}_2 = C_2(\hat{\mathcal{X}})$            ▷ Query classifier
7:    $W_1 : T(\mathcal{M}, \mathcal{Z}_1), W_2 : T(\mathcal{M}, \mathcal{Z}_2)$      ▷ Train local regression model
8:    **if** Unknown relation between $D_1$ and $D_2$ **then**
9:      **for** $i \in D_1$ **do**
10:        $W_2[i] = W_2\left[\arg\min_{j \in D_2}\left(\|W_1[i] - W_2[j]\|\right)\right]$    ▷ Class alignment
11:    $\mathcal{S}_1$.append$(W_1), \mathcal{S}_2$.append$(W_2)$         ▷ Create model signature
12: $d = Distance(\mathcal{S}_1, \mathcal{S}_2)$               ▷ Compute model distance

---

## 4 VALIDATION

### 4.1 IMPLEMENTATION DETAILS

We validate our proposed method, Zest, in the vision, text, and audio domains using four publicly available classifiers and four datasets: ResNet20 trained on CIFAR-10, ResNet50 trained on CIFAR-

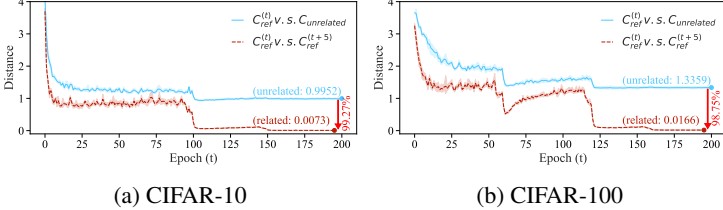

(a) CIFAR-10          (b) CIFAR-100

Figure 2: The distance computed by Zest for unrelated classifiers ($C_{\text{ref}}^{(t)}$ versus $C_{\text{unrelated}}$) and related classifiers ($C_{\text{ref}}^{(t)}$ versus $C_{\text{ref}}^{(t+k)}$) as a function of training epoch, $t$. We consider $k = 5$ for the related classifier. See Appendix D for the results of other $k$ values. The distance of related classifiers is significantly smaller than the Zest distance of unrelated classifiers, especially as training converges.

100, LSTM trained on AG News, and M5 network trained on Speech Commands. The accuracy of ResNet20 on CIFAR-10 is $91.77(\pm 0.17)\%$, $76.17(\pm 0.22)\%$ for the ResNet50 on CIFAR-100, $90.27(\pm 0.52)\%$ for LSTM on AG News, and $87.67(\pm 0.86)\%$ for M5 network on Speech Commands. Our results in this section are averaged over 5 different training runs of classifiers under comparison. Recall from § 3 that Zest consists of 4 successive phases. In phase 1 of Zest, we select 128 samples randomly from the training set. In phase 1 and phase 2, we follow the same implementation and setup as the original LIME paper. In phase 3, we train linear regression models using the least-squares estimation technique. We use relative Cosine similarity (see Appendix C for details) for computing the distance between global signatures in phase 4. However, in Appendix G we perform an ablation study where we compare using the $\ell_1$ and $\ell_2$ norms as alternative metrics, and show that Zest is agnostic to the choice of metric used to compare global signatures.

## 4.2   DOES ZEST CAPTURE THE DISTANCE BETWEEN CLASSIFIERS CORRECTLY?

We analyse the performance of Zest in computing distances between *related* or *unrelated classifiers*. To build related and unrelated pairs, we first collect a reference classifier trained with $t$ epochs, $C_{\text{ref}}^{(t)}$. We consider the classifier at epoch $t+k$ as a related classifier, $C_{\text{ref}}^{(t+k)}$, to the reference classifier $C_{\text{ref}}^{(t)}$. Instead, we consider the same architecture but trained with a different random seed as a classifier, $C_{\text{unrelated}}^{(t)}$, unrelated to the reference classifier. The distance between unrelated classifiers ($C_{\text{ref}}^{(t)}$ versus $C_{\text{unrelated}}$) should be bigger than the distance between related classifiers ($C_{\text{ref}}^{(t)}$ versus $C_{\text{ref}}^{(t+k)}$).

Figure 2 shows the Zest distance between unrelated and related classifiers, at training epochs $[0, 200]$ of the reference classifier. Overall, model distances in both the unrelated and related cases decrease as training progresses and model weights become more informative. This is intuitive given that both classifiers are trained to solve the same task. However, the distance output by Zest for unrelated classifiers has a lower bound that is significantly higher than for related classifiers. For example, the distance computed by Zest for unrelated ResNet20 classifiers is still above 1 at the last epoch (epoch=200), while it is so close to 0 for related ones with $k = 5$. Therefore, **Zest can correctly distinguish related from unrelated classifiers.**

## 4.3   HOW SENSITIVE IS ZEST TO THE SIZE OF REFERENCE SET?

We analyse the effect of the reference set on the performance of Zest by studying 1) the effect of the size of the reference set on Zest; 2) the effect of stochasticity in the selection of points that make up the reference set. For both of these analyses, we consider two pairs of related classifiers: ($C_{\text{ref}}^{(100)}$ versus $C_{\text{ref}}^{(200)}$) and ($C_{\text{ref}}^{(190)}$ versus $C_{\text{ref}}^{(200)}$). Figure 3 shows the average and standard deviation of the distance computed by Zest for different sizes, $\{1, 2, 4, 8, 16, 32, 64, 128\}$, of the reference set and different random samples of reference points. The confidence interval (coloured areas) is obtained by repeating the experiment 5 times to train classifiers with different seeds. We observe that the average distance computed by Zest is identical regardless of the size of the reference set in both of these related classifiers. However, the standard deviation of the distance computed by Zest decreases as the size of the reference set increases: for instance, in $C_{\text{ref}}^{(100)}$ versus $C_{\text{ref}}^{(200)}$ standard deviation decreases from 0.3 for a reference set of size 1 to a value close to zero as we increase the reference size to 128. Note that this remains a small reference set relative to the size of the training set used to learn these classifiers. In the case of $C_{\text{ref}}^{(190)}$ versus $C_{\text{ref}}^{(200)}$, the classifiers are so similar in the first place that the standard deviation of the distance computed by Zest is close to zero and remains similar for all of the reference set sizes.

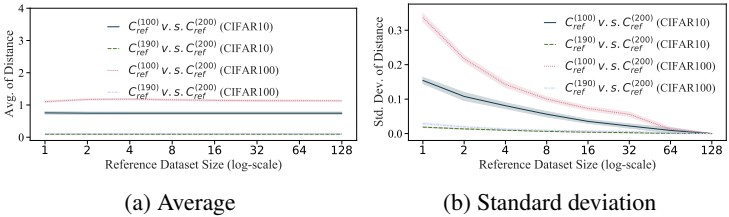

(a) Average    (b) Standard deviation

Figure 3: Influence of the reference set size and sampling stochasticity on the distance computed by Zest. The average Zest distances of different reference sets are similar, while the standard deviation decreases as the size of the reference set increases.

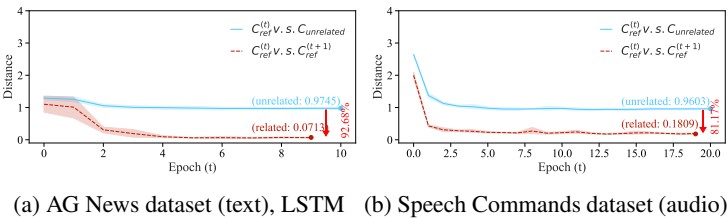

(a) AG News dataset (text), LSTM    (b) Speech Commands dataset (audio), M5

Figure 4: Evaluation of correctness of Zest in the text and audio domains respectively: this figure is plotted in a similar manner to Figure 2, *i.e.,* the distances between related and unrelated classifier pairs are plotted as a function of training epoch, $t$. We can see the Zest distance of related classifiers are significantly smaller than distances of unrelated classifiers across domains.

Such a quick reduction in variance by increasing the number of reference points can not be achieved by prediction comparison. In the same training setting for CIFAR-10 and using 128 reference points, the prediction similarity between $C_{\text{ref}}^{(100)}$ and $C_{\text{ref}}^{(200)}$ is $93.28(\pm 1.06)\%$, which is overlapping with the prediction similarity between 2 unrelated models: $91.52(\pm 2.27)\%$. The two values become $92.27(\pm 0.22)\%$ and $91.59(\pm 0.76)\%$ respectively when we increase the reference dataset to size 1280. Such a small gap and large variance make it hard to confidently claim if two models are the same and can lead to a high false positive rate (*i.e.,* claiming two unrelated models as related). This is opposite to distance computed by Zest on a reference dataset that is 10 times smaller, which can easily separate distances between unrelated and related pairs, as shown in Figure 2. One may think the bad performance of prediction comparison is because most of data points are far away from the decision boundaries of the well-trained models, and decide to pick data points near the decision boundaries. By doing so, we observed a similarity of $35.23(\pm 2.60)\%$ between unrelated models and $41.92(\pm 2.82)\%$ between related models. However, such points are sensitive to small changes in the decision boundaries. By training a model to match the predictions of the victim model (can be thought of as high-fidelity model extraction attacks), we find the similarity between the victim and extracted models quickly drops to $32.36(\pm 1.84)\%$, being no different from the similarity between two unrelated models. This problem is solved by Zest, as described in details in § 5.2.

### 4.4    Zest in other domains

In this section, we analyse the performance of Zest in computing distances between related and unrelated model pairs in both text and audio domains. In the text domain we used Long-Short Term Memory (LSTM) classifiers (Hochreiter & Schmidhuber, 1997) and AG News dataset (Zhang et al., 2015) (described in Appendix A.1 and A.2). In the audio domain we used M5 speech classifier (Dai et al., 2017) and Speech Commands dataset (Warden, 2018). Similarly to our vision experiment in Section 4.2, related LSTM (or M5) classifiers are checkpoints at two different epochs, while unrelated LSTM (or M5) classifiers are trained with different random seeds. Figure 4 shows that the distance output by Zest for unrelated classifiers in both text and audio domains is significantly higher than for related classifiers (consistent with our results in Section 4.2 on vision datasets). This thus reinforces that Zest is applicable to different domains of vision, audio and text.

## 5    Case Studies: Applications of Zest

Next, we evaluate the applicability of distances between reference and suspicious classifiers, computed by Zest to two tasks (see Table 2 in Appendix A.3). Our case studies, introduced earlier, are model stealing and machine unlearning. We first introduce relevant literature on these case studies.

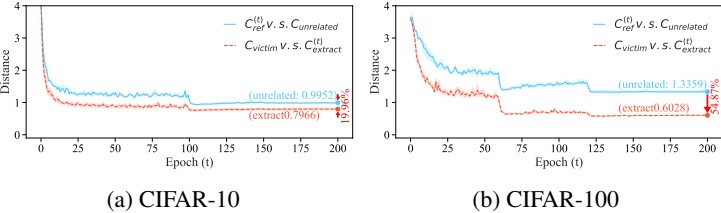

(a) CIFAR-10            (b) CIFAR-100

Figure 5: The distance between a victim classifier and its extracted copy obtained by Zest as a function of training epochs of the latter. Zest can detect the extracted copy of the victim classifier.

## 5.1 BACKGROUND ON MODEL STEALING AND MACHINE UNLEARNING

**Model stealing.** There is often a strong need to protect the confidentiality of ML models; it can be costly to collect datasets and train models, making them valuable intellectual property. Model stealing refers to an attacker duplicating the functionality of a confidential model without permission, stealing intellectual property. Model extraction is a class of such attacks that efficiently learn the approximate functionality of a model by observing its outputs on selected inputs (Tramèr et al., 2016). This is often achieved via an exposed inference API, which is common in ML-as-a-service.

Model stealing can also be considered a malicious case of model reuse, where a model is used as a basis for training another model at a smaller cost (e.g., through fine-tuning). Li et al. (2021) developed a distance metric, ModelDiff, that detects model reuse in DNNs by comparing the models' outputs. However, while Li et al. (2021) identified model reuse via transfer learning and model compression, they were unsuccessful in detecting models stolen via model extraction, and presented this as an open problem. In our case study, we show we are able to fill this gap in the literature.

**Machine unlearning.** Private data is often collected for the purpose of training ML models, but legislation and privacy ethics call for the right to be forgotten. For ML models, the right to be forgotten implies that a user should have the right to remove the effects of their data on a model trained on that data. The naive solution is to remove the user's entry from training data, then retrain the model, but there are more efficient methods: these either decrease the time it takes to exactly/directly retrain (Bourtoule et al., 2021) or find a way to approximate the retraining procedure (Graves et al., 2020). After requesting data removal, the user (or third-party auditors) may be interested in verifying that the new model has indeed unlearned their data. By measuring the similarity of the decision boundary between the old and new models around the user's data, we can use Zest as a heuristic to help test if the data has been unlearned.

## 5.2 ZEST DETECTS MODEL EXTRACTION

We now aim to compute the distance between a reference victim classifier and a copy of the victim classifier extracted by an adversary. We consider a strong adversary who knows the architecture of the victim model and has access to the unlabeled training set of the victim classifier. This makes it easier for the adversary to train a copy of the model which is hard to distinguish from the victim model they are stealing. The adversary trains the same architecture classifier using the training set and their predicted labels obtained through querying the victim model.

In Figure 5, we use Zest to compute the distance between the victim classifier and the extracted classifier as a function of training epochs of the extracted classifier. As the accuracy of the extracted classifier improves during its training (reflected by the increasing number of training epochs), the distance between the victim and extracted classifiers decreases for both CIFAR-10 and CIFAR-100. The minimum distances achieved at the last epoch are 0.80 and 0.60 and remain smaller than the lower bound of unrelated classifiers (discussed in § 4.2). Therefore, **Zest can help detect a copy extracted with the attack considered here based on its distance to the victim classifier**.

To further our evaluation, we also compare our proposed method Zest using the ModelReuse benchmark introduced with ModelDiff (Li et al., 2021). The benchmark contains 144 pairs of reused models which contain one model that reuses the knowledge of another model. Here, Li et al. (2021) define reusing as transfer learning, pruning, quantization, distillation, model extraction and their combinations (see Table 2 in Li et al. (2021) for the complete description of these tasks). Thus, this benchmark not only allows us to compare the performance of Zest with ModelDiff when it comes to detecting model extraction but also these several other forms of model reuse. Following the setting of ModelDiff's evaluation, we use ResNet18 (He et al., 2016) and MobileNet-v2 (Sandler et al., 2018) trained on ImageNet (Deng et al., 2009), Flower102 (Nilsback & Zisserman, 2008), or

| Reuse Task | #Models | Zest (Cosine) | ModelDiff |
|---|---|---|---|
| Transfer | 12 | 100.0% | 100.0% |
| Prune | 36 | 100.0% | 100.0% |
| Quantization | 12 | 100.0% | 100.0% |
| Distillation | 12 | 100.0% | 100.0% |
| Model extraction | 12 | 100.0% | *0.0 %* |
| Transfer + prune | 36 | 100.0% | 100.0% |
| Transfer + quantize | 12 | 100.0% | 100.0% |
| Transfer + distill | 12 | 100.0% | 100.0% |
| Overall | 144 | **100.0**% | 91.7% |

Table 1: Accuracy of our proposed method, Zest, and ModelDiff (Li et al., 2021) in detecting the similarity of reused model pairs in ModelReuse benchmark. The maximum of overall accuracy is highlighted by bold, while the minimum of accuracy among all the task is highlighted by italic.

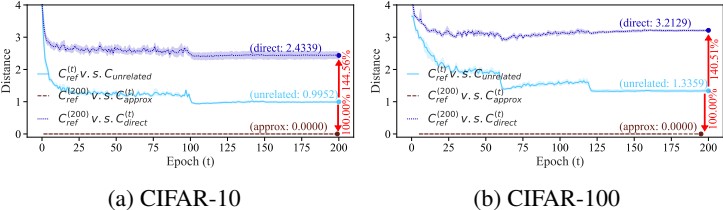

(a) CIFAR-10                    (b) CIFAR-100

Figure 6: Distances between the original classifier and (1) a directly unlearned classifier, (2) an approximately unlearned classifier when data to be unlearned serves as the reference set in Zest.

SDog120 (Khosla et al., 2011) datasets. To compute the accuracy of Zest and ModelDiff, we first use Zest to compute distances between models in each pair (reported in Appendix F). It is worth to note that there are model pairs trained on different dataset in this benchmark, which is where class alignment comes to use. Then, we threshold the normalized distance obtained by Zest: distances lower than 1 indicate similar models while those higher than 1 indicate dissimilar models. Table 1 shows that Zest and ModelDiff have similar performance in all tasks except model extraction in which the accuracy of ModelDiff is zero while the accuracy of Zest is 100%. Therefore, **Zest outperforms ModelDiff in the ModelReuse benchmark's model extraction task.**

### 5.3 ZEST VERIFIES UNLEARNING

In machine unlearning, our reference classifier is the original classifier trained on all the training set. We compare two classifiers to this reference classifier. The first is the exactly/directly unlearned classifier, i.e., a classifier trained on all the training sets except data that was requested to be unlearned as done in Bourtoule et al. (2021). The second is the approximately unlearned classifier, i.e., a classifier obtained by directly adding an update to the original reference classifier with the goal of removing the effect of data to be unlearned as done in Graves et al. (2020).

We consider 128 data points to be unlearned, and choose them as the reference set for training the LIME linear regression models which underlie our distance metric. In Figure 6, we compute model distances between the three classifiers mentioned above. We observe that the distance computed by Zest between the reference and exactly unlearned classifiers (or approximately unlearned classifier) is high, even higher than the distance between unrelated classifiers (discussed in § 4.2). However, the distance between the reference and approximately unlearned classifiers is almost null, confirming that these techniques cannot completely remove the influence of data requested to be unlearned from the model. Therefore, **Zest suggests that exact unlearning can be distinguished from approximate unlearning (or not unlearning) by computing model distances**.

## 6 CONCLUSION

We proposed Zest, an approach to compute architecture-independent model distances to measure similarity between ML models. Zest only requires black-box access to the compared models. Zest achieves this by learning linear models with LIME to capture local behavior, then concatenating their weights for an approximation of the compared models' global behaviors. To compute a distance for the compared models, we take the Cosine distance between the weights of their concatenated linear models. We demonstrated the utility of Zest on several tasks, including many common forms of model reuse, and machine unlearning. We expect that future work will improve upon Zest by better understanding how to approximate locally the behavior of models, and understanding how to compare local approximations to provide finer-grained insights into model similarities and differences.

ETHICAL IMPACT STATEMENT

Our work applies to two tasks, the detection of model stealing and verification of machine unlearning, that have practical impact to the protection of intellectual property and user privacy. Model stealing is theft on intellectual property, and the stolen model can be analyzed to extract private data used in training; machine unlearning enforces users' right to be forgotten. We believe that research on model distance metrics can be used to help deter model stealing, and help hold model owners responsible to ethical codes and regulations that call for the right to be forgotten, such as the General Data Protection Regulation in the European Union, the California Consumer Privacy Act in the United States, and the Personal Information Protection and Electronic Documents Act in Canada.

REPRODUCIBILITY STATEMENT

To share our code for reproducibility during the review process, we will provide a link to an anonymous repository in the discussion forums. In the future, we also aim to publicly open-source our code base, and provide documentations for reproducing the results shown in this paper. When possible, we used well-established, open-source implementations of models and algorithms to further decrease the overhead of reproducing our experiments. We provide the complete list of code repositories that we referenced in Appendix E.

ACKNOWLEDGMENTS

We would like to acknowledge our sponsors, who support our research with financial and in-kind contributions: CIFAR through the Canada CIFAR AI Chair and AI Catalyst program, DARPA through the GARD program, Intel, Meta, NFRF through an Exploration grant, and NSERC through the Discovery Grant and COHESA Strategic Alliance. Resources used in preparing this research were provided, in part, by the Province of Ontario, the Government of Canada through CIFAR, and companies sponsoring the Vector Institute. We would like to thank members of the CleverHans Lab for their feedback.

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

## A  DATASETS, CLASSIFIERS AND TASKS

### A.1  DATASETS

We evaluated Zest in seven different datasets. Five of these are vision datasets–CIFAR-10, CIFAR-100, ImageNet (Deng et al., 2009), Flower102 (Nilsback & Zisserman, 2008), and SDog120 (Khosla et al., 2011); the other two are a text dataset– AG News (Zhang et al., 2015), and an audio dataset– Speech Commands (Warden, 2018). Below, we provide a brief description of each dataset.

**CIFAR-10 and CIFAR-100**  are widely used small image classification datasets. Both datasets contain 60,000 32x32 pixel colour images. CIFAR-10 has 10 classes, each with 6,000 images, and CIFAR-100 has 100 classes, each with 600 images. The images include common animals and man-made objects, like cats, deers and ships.

**ImageNet**  is a large vision dataset for image classification with 1,000 classes. It contains 1.4 million labeled images of varying sizes, with an average resolution of 469x387 pixels. The images include a wide variety of organisms and man-made objects, including different species of flowers and dogs. In the ModelReuse dataset, the knowledge of models trained on ImageNet are reused to label data for the Flower102 and SDog120 datasets.

**Flower102**   is an image classification dataset that contains 102 classes of flowers. Each class has between 40 to 258 samples of varying sizes, for a total of 8,189 images.

**SDog120**   is the 120 class Stanford Dogs dataset, an image classification dataset that contains 22,000 labeled dog images. Each class contains 150-200 samples of varying sizes larger than 200x200 pixels.

**AG News**   is a text dataset for topic classification. It is a curated subset of the AG News Corpus, and contains 127,600 news articles in 4 topic classes, collected from web sources; each class contains 31,900 samples.

**Speech Commands**   is an audio dataset of spoken words for speech recognition. It contains 105,829 utterances of 35 words, performed by 2,618 speakers; each word is a class.

## A.2   CLASSIFIERS

For the architectures of the classifiers, we used ResNet20 (He et al., 2016), ResNet50 (He et al., 2016), ResNet18 (He et al., 2016), MobileNet-v2 (Sandler et al., 2018), LSTM (Hochreiter & Schmidhuber, 1997) and M5 (Dai et al., 2017).

In the vision domain, we used ResNet20 trained on CIFAR-10, and ResNet50 trained on CIFAR-100. The ModelReuse benchmark we used contained ResNet18 and MobileNet-v2 pre-trained on ImageNet, then reused for Flower102 and SDog120.

For the text domain, we used a Long Short-Term Memory (LSTM) model trained on the AG News dataset. For the audio domain, we used the an M5 model trained on the Speech Commands dataset.

We used publicly available implementations of all the classifiers, and include the code source for each of the them in Appendix E.

## A.3   SETTINGS FOR CASE STUDIES OF ZEST

| Task | Reference classifier | Suspicious classifier |
|---|---|---|
| Model extraction | Victim classifier | Extracted copy of the victim classifier |
| Machine unlearning | Original classifier | Directly or Approximately unlearned classifier |

Table 2: Settings of our case studies.

## A.4   TASKS IN MODELREUSE

We provide a brief description of the reuse tasks in the ModelReuse benchmark, taken as excerpts from Li et al. (2021). Each of these tasks start with a ResNet18 or MobileNetV2 model trained on Imagenet, then reuses the knowledge in some way to classify data points from the Flower102 and SDog120 datasets, as described in Table 3. See Table 5 for our complete results on the ModelReuse benchmark.

| Reuse Task | Description |
|---|---|
| Transfer (tune $X\%$) | Transfer each source model to each target dataset, fine-tune the last $X\%$ layers. |
| Prune ($X\%$) | Prune $X\%$ weights in each transferred model and fine-tune. |
| Quantize | Compress each transferred model using post-training weight quantization to the int8 data type. |
| Distill | Distill each transferred model to a target model with the same architecture using feature distillation. |
| Steal | Use the output of each transferred model to train a target model with different architecture. |

Table 3: The reuse method used for each task in the ModelReuse benchmark.

## B  CLASS ALIGNMENT

We evaluate our class alignment strategy in two different settings of comparing two classifiers $C_1$ and $C_2$ (we assume the classifiers have performance better than random guessing):

1. Both models are trained with the same architecture and on the same dataset but with different output label orders;

2. Each model is trained with the same architecture on a different dataset and with a different number of labels;

In setting (1), $C_1$: is trained on CIFAR10 (or CIFAR100) and $C_2$: trained on CIFAR10 (CIFAR100) but with randomly swapped labels. Our experimental results show that our class alignment strategy is always able to correctly match the classes of two models in this setting. (e.g. matching the "cat" classes of the two models when the "cat" class of one of the models is indexed by 3 while the "cat" class in the other model is indexed by 7). The intuition behind this is that class alignment finds the class pairs with the most similar LIME masks, in other words, the class predictions are made using similar features of the inputs.

Setting (2) includes the transfer learning task of ModelReuse experiments. In this setting, models are trained on two different datasets with a different number of labels (1000 classes for ImageNet versus 120 for SDog120, or 1000 classes for ImageNet versus 102 for Flower102). These two datasets however share some common classes such as breeds of dog or type of flowers. For example in one of the experiments, $C_1$ is pre-trained on ImageNet with 1000 classes and $C_2$ is fine-tuned $C_1$ on Flower102 with 102 classes. The class alignment of these two classifiers with unrelated labels are as follows:

1. Train LIME linear models of $C_1$ using 128 reference data points, and train LIME linear models of $C_2$ using 128 reference data points.

2. Concatenate weights of all LIME linear models of $C_1$ class-wise to obtain $S_1 = \{W_1^{p_1}|...|W_1^{p_{128}}\}$ as the order of labels are fixed, and concatenate weights of all LIME linear models of $C_2$ class-wise to obtain $S_2 = \{W_2^{p_1}|...|W_2^{p_{128}}\}$.

3. Setting a reference classifier: we choose the classifier $C_2$ that includes the lower number of classes (102 labels) as a reference classifier;

4. Compute the Cosine distance between each row of $S_2$ with all rows of $S_1$. We do this because we assume that the semantic of labels and number of common labels are unknown.

5. Pick 102 pair rows with the minimum distances, determining 102 classes chosen out of 1000 classes of $C_1$. This helps us to compare $C_1$ and $C_2$ based on the classes that perform most similarly to each other. This means that we might have pairs of aligned classes that are not related semantically, but this is expected and will contribute to telling the classifiers apart distance-wise.

In another example of the ModelReuse transfer learning experiments, we compare a ResNet18 trained on SDog120 with another ResNet18 trained on ImageNet. As mentioned in Appendix A.1, SDog120 contains 120 breeds of dogs. On the other hand, ImageNet includes 130 breeds of dogs out of its 1000 classes. In this experiment, our class alignment is able to pick 120 similar breeds of dog from ImageNet which are most similar to the 120 breeds of dogs from SDog120.

## C  RELATIVE DISTANCE

We use both unbounded distance metrics and bounded distance. For better visualisation and fairer comparison, we scale all of them similarly. In particular, we scale the Zest distance of the compared models by the average Zest distance of unrelated models on similar tasks. Note that the unrelated models are obtained using different seeds. We define the relative distance as the ratio between the Zest distance of the compared models and the average Zest distance of multiple unrelated models in the last epoch:

$$\text{Relative distance} = \frac{\text{Computed model distance}}{\text{Average distance of multiple unrelated models}}. \qquad (1)$$

# D   ABLATION STUDY ON THE VALUE OF $K$

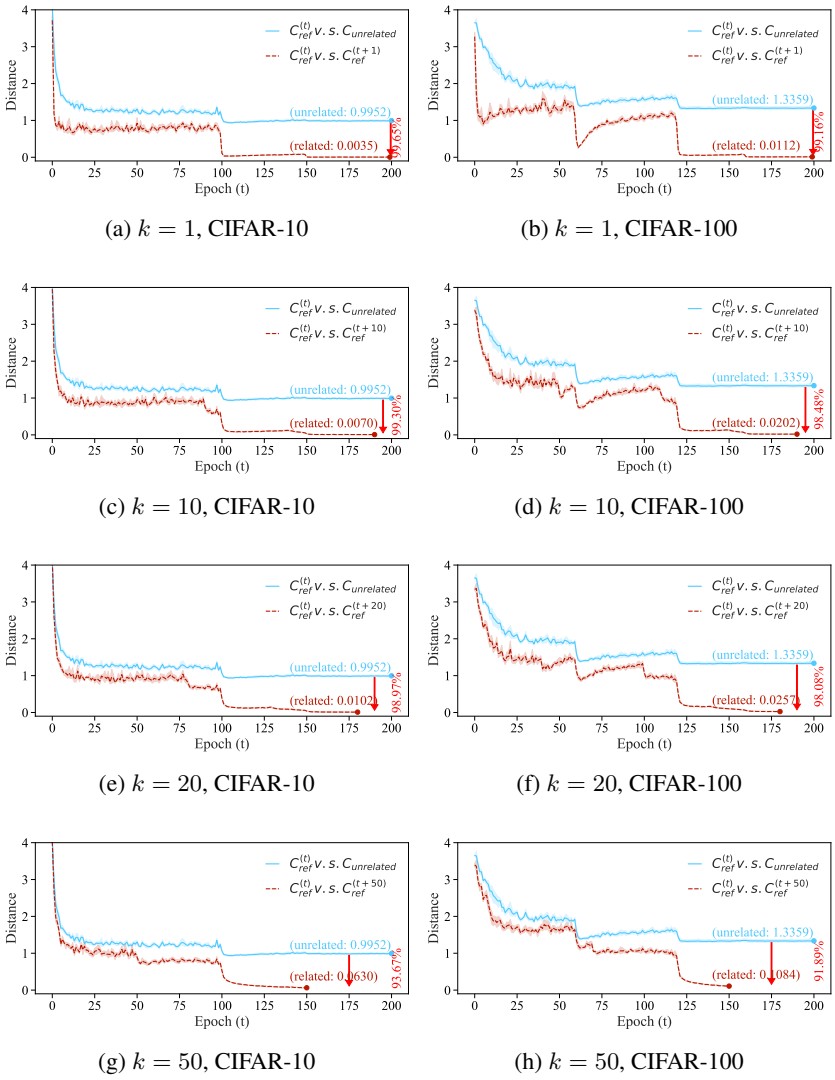

Figure 7: Impact of the value of k on the Zest distances between related models: we vary the value of k and plot the distances between a model at training epoch $t$ and itself after training for another $k$ epochs for CIFAR-10 and CIFAR-100 datasets respectively.

In this section, we perform an ablation study on the value of $k$ for both CIFAR-10 and CIFAR-100. Figure 7 shows the Zest distances between related models (i.e. the same model's checkpoints at epoch $t$ and epoch $t + k$, where $k$ is set to $1, 10, 20, 50$ respectively), and between unrelated models. We can observe that

1. The Zest distances between related models obtained with different k values are significantly smaller than the Zest distances between unrelated models;

2. The larger the value of k, the larger the Zest distance between related models. For example, the Zest distance between related CIFAR-10 models for k values of 1, 10, 20 and 50 are 0.0035, 0.0070, 0.0120 and 0.0630, respectively. Remember from Section 4.2 that each related pair contains a reference classifier trained with $t$ epochs and another classifier at epoch $t + k$. Therefore, increasing the value of k results in more changes in the parameters of the model over k epochs that are captured by our proposed method Zest.

## E    EXTERNAL CODE SOURCES

We used open source code in our experiments to implement our ResNet-20 and ResNet-50 models, as well as the LIME algorithm. We also obtained the ModelReuse benchmark Li et al. (2021) from the authors' repository. We list the URLs of the public code repositories we referenced in Table 4.

| Task | Code Source |
|---|---|
| ResNet-20 on CIFAR-10 | `github.com/akamaster/pytorch_resnet_cifar10` |
| ResNet-50 on CIFAR-100 | `github.com/weiaicunzai/pytorch-cifar100` |
| LIME | `github.com/marcotcr/lime` |
| ModelReuse benchmark | `github.com/ylimit/ModelDiff` |
| LSTM on AG News | `pytorch.org/tutorials/beginner/text_sentiment_ngrams_tutorial.html` |
| M5 on Speech Commands | `pytorch.org/tutorials/intermediate/speech_command_recognition_with_torchaudio.html` |

Table 4: Sources of codes used for experiments from external

## F    DISTANCE OF CLASSIFIERS IN MODELREUSE

| Reuse Task | #Models | $\ell_2$ Distance | Acc. | $cos$ Distance | Acc. |
|---|---|---|---|---|---|
| Transfer (tune 10%) | 4 | 0.745(±0.075) | 100.0% | 0.816(±0.040) | 100.0% |
| Transfer (tune 50%) | 4 | 0.682(±0.185) | 100.0% | 0.843(±0.041) | 100.0% |
| Transfer (tune 100%) | 4 | 0.703(±0.207) | 100.0% | 0.851(±0.037) | 100.0% |
| Prune 20% | 12 | 0.421(±0.143) | 100.0% | 0.218(±0.141) | 100.0% |
| Prune 50% | 12 | 0.476(±0.128) | 100.0% | 0.276(±0.137) | 100.0% |
| Prune 80% | 12 | 0.829(±0.304) | 58.33% | 0.555(±0.229) | 100.0% |
| Quantize | 12 | 0.074(±0.155) | 100.0% | 0.067(±0.150) | 100.0% |
| Distill | 12 | 0.633(±0.136) | 100.0% | 0.571(±0.166) | 100.0% |
| Steal | 12 | 0.716(±0.130) | 100.0% | 0.752(±0.131) | 100.0% |
| Transfer + prune 20% | 12 | 0.692(±0.225) | 100.0% | 0.861(±0.040) | 100.0% |
| Transfer + prune 50% | 12 | 0.715(±0.217) | 91.67% | 0.856(±0.046) | 100.0% |
| Transfer + prune 80% | 12 | 0.893(±0.241) | 75.00% | 0.868(±0.038) | 100.0% |
| Transfer + quantize | 12 | 0.707(±0.171) | 100.0% | 0.840(±0.045) | 100.0% |
| Transfer + distill | 12 | 0.514(±0.127) | 100.0% | 0.909(±0.060) | 100.0% |
| **Overall** | 144 | 0.615(±0.282) | 93.75% | 0.634(±0.308) | 100.0% |

Table 5: The comparison of Zest with ModelDiff on ModelReuse benchmark.

We provide a more complete results table for our experiments on the ModelReuse benchmark. Table 1 provides more granularity on the reuse tasks, breaking each task into subtasks (e.g. transfer is separated to three subtasks based on different levels of tuning). We also include the average distance values and standard deviations, and show the results of using the $\ell_2$ norm to measure the distance between the collections of linear approximations of the compared models. The $\ell_2$ norm is significantly more sensitive to pruning: the accuracy of Zest decreases significantly in the Prune 80% and Transfer + prune 50% or 80% tasks. In comparison, by using the cosine distance, Zest detects every reuse task across the board.

## G    ABLATION STUDY ON THE CHOICE OF METRIC

We provide additional experiments on comparing Zest's performance when using different metrics to compute the distance between the global signatures. This is an extension of our experiments in § 4. Specifically, we repeat the experiments shown in Figures 2, 5 and 6 using the $\ell_1$ and $\ell 2$ norms as distance metrics. The respective experiments are distinguishing related and unrelated classifiers, detecting model extraction, and verifying machine unlearning. Our results show that for all tasks, Zest can also capture the general similarity between compared models using $\ell_1$ and $\ell 2$ distances

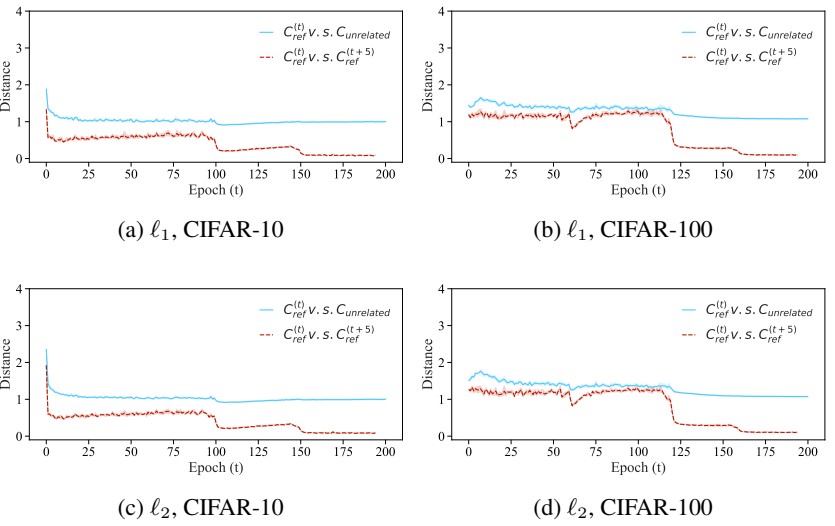

Figure 8: Re-plotting Figure 2 using $\ell_1$ and $\ell_2$ distances.

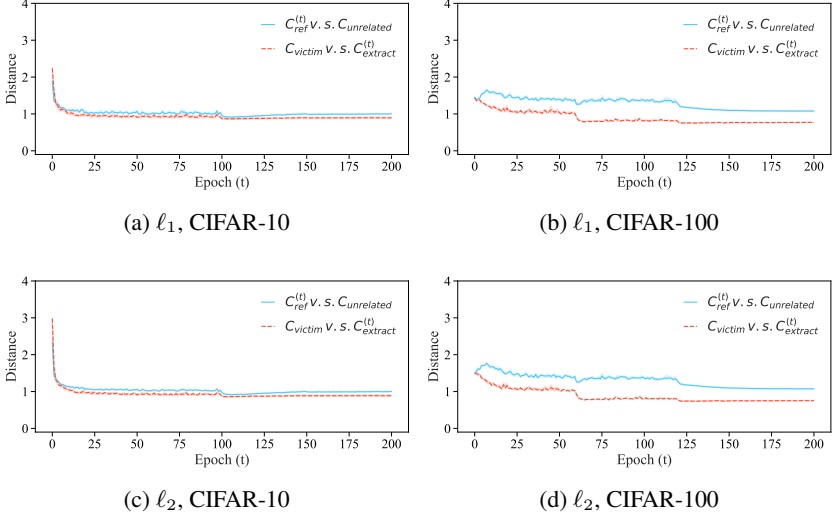

Figure 9: Re-plotting Figure 5 using $\ell_1$ and $\ell_2$ distances.

instead of the Cosine distance. We used the Cosine distance for our main experiments because it performs better on average.

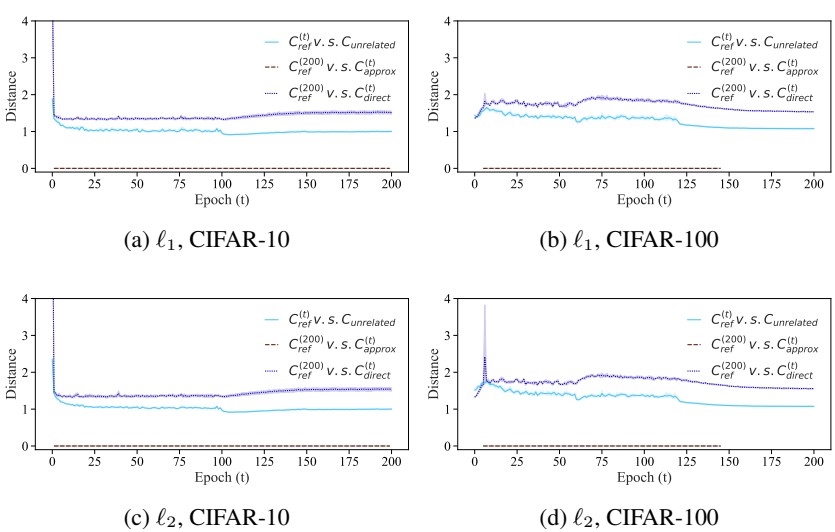

Figure 10: Re-plotting Figure 6 using $\ell_1$ and $\ell_2$ distances.

