# OpenReview forum: "A Zest of LIME: Towards Architecture-Independent Model Distances"
_ICLR.cc/2022/Conference — ICLR 2022 Poster_

### Official Review · Reviewer_GFcZ · 2021-10-24

**Correctness:** 4
**Technical Novelty And Significance:** 2
**Empirical Novelty And Significance:** 3
**Recommendation:** 6
**Confidence:** 4

**Main Review:**

This paper utilizes a model explanation methods called LIME to design a new method to calculating the distance between two machine learning models. Overall, the problem is well motivated by stating the limitations of current approaches, and the proposed method is clearly presented with experiments to support its benefits:

1. The authors clearly states the limitations of existing approaches, namely architecture dependent or dependence on the choice of model inputs. The proposed method interpolates between two extremes and thus alleviate both drawbacks.

2. The authors provide a thorough review  related work, and introduce the proposed method clearly.

3. The experiments are carefully designed to demonstrate the benefits of the proposed method.

In general, I have the following several concerns:

1. Algorithm 1 line 9 - 10, or in particular the way to deal with class alignment. I feel its a bit unclear: looks like we care finding the minimum alignment from all possible alignments. But if it actually due to class mismatch, is it possible that we get a small distance while the models are actually different? For example, model 1 predicts class A while model 2 predict class B using similar weights.

2. Algorithm 1 line 12, the distance is chosen to be a cosine operation, so why it will be larger than 1 in some of the experiments?

3. Section 4.2, any reason to consider k = 5? It might be helpful to include additional experiment results varying the values of k.

4. For the applications mentioned, I cannot help my self thinking of knowledge distillation, which seems to be closely related to model stealing. Is there any related work discussing this or even further, can ZEST find application for knowledge distillations?

**Summary Of The Paper:**

The paper proposes a new method to calculate the distance between two machine learning models. By comparing using local approximations (LIME) and compute the Cosine distance, this yields an architecture-independent method operating in weight space. It has proves effectiveness in applications including detecting model stealing and verifying machine unlearning.

**Summary Of The Review:**

Overall I think this is a good paper and can be accepted to ICLR. Although it does not have much theoretical novelty, I believe applying LIME in the area of calculating distance between two models is pretty new. If my listed concerns can be addressed, I willing to increase my ratings one level further.

---

> ### Author Response · Authors · 2021-11-20
> **Response 1**
>
> We thank the reviewer for the comments and suggestions. In the text below we respond to the comments.
>
> >_**Algorithm 1 line 9 - 10, or in particular the way to deal with class alignment. I feel it's a bit unclear: looks like we care about finding the minimum alignment from all possible alignments. But if it is actually due to class mismatch, is it possible that we get a small distance while the models are actually different? For example, model 1 predicts class A while model 2 predicts class B using similar weights.**_
>
> The reason we conduct class alignment this way is because in some applications such as transfer learning, two classifiers ($C_1$ and $C_2$) under evaluation have different classes and numbers of classes. For example, in one of our transfer learning experiments: $C_1$ is a pre-trained classifier on ImageNet with 1000 classes, and $C_2$ is obtained by fine-tuning this pre-trained model on Flower102 with 102 classes.
>
> To compute the distance between $C_1$ and $C_2$, we find the 102 classes of $C_1$ that are most similar to the classes of $C_2$, pairing the similar classes together, and compare the similarity between the classifiers' decision boundaries for these classes. In general, to compare two models with different classes (that could have different numbers of classes), we compare their behavior on the top $n$ most similar classes, where $n$ is the number of classes of the model with fewer classes. To find similar classes, we compare the models' LIME explanations: two classes are similar if their explanation (i.e. LIME masks for the 128 data points used in the reference set) are similar.
>
> Since both class alignment and the Zest distance are computed over the entire reference set, it is unlikely that two semantically unrelated classes will have a small distance, because that requires the models to share similar explanations (on average) over 128 data points.
>
> We also added additional details on our class alignment method, including a formal description of how we find the most similar classes, in Appendix B.
>
>
> >_**Algorithm 1 line 12, the distance is chosen to be a cosine operation, so why it will be larger than 1 in some of the experiments?**_
>
> The Zest distances (e.g. in Figure 2) are sometimes larger than 1 because we report relative distance instead of actual distance. We use both unbounded distance metrics ($l_1$ and $l_2$, see Appendix G) and bounded Cosine distance. For better visualisation and fairer comparison, we scale all of them similarly. In particular, we scale the Zest distances based on the distances of the unrelated models. To make this clear, we detail the description of our scaling in Appendix C:
>
> “
> We use both unbounded distance metrics and bounded distance. For better visualisation and fairer  comparison, we scale all of them similarly. In particular, we scale the Zest distance of the compared models by the average Zest distance of unrelated models on similar tasks. Note that the unrelated models are obtained using different seeds. We define the relative distance as the ratio between the Zest distance of the compared models and the average Zest distance of multiple unrelated models in the last epoch:
> \begin{equation}
> 	\text{Relative distance}=\frac{\text{Computed model distance}}{\text{Average distance of multiple unrelated models}}.
> \end{equation}
> ”
>
> >_**Section 4.2, any reason to consider k = 5? It might be helpful to include additional experiment results varying the values of k.**_
>
> Due to the page limit, we only added the results of k=5 at the time of submission. In response to your suggestion, we added a new Appendix D in the manuscript to show the results for other values of k (k = 1, 10, 20, 50):
>
> “
> In this section, we perform an ablation study on the value of $k$ for both CIFAR-10 and CIFAR-100. Figure 7 shows the Zest distances between related models (i.e. the same model's checkpoints at epoch $t$ and epoch $t+k$, where $k$ is set to $1, 10, 20, 50$ respectively), and between unrelated models. We can observe that
> 1. The Zest distances between related models obtained with different k values are significantly smaller than the Zest distances between unrelated models.
> 2. The larger the value of k, the larger the Zest distance between related models. For example, the Zest distance between related CIFAR-10 models for k values of 1, 10, 20 and 50 are 0.0035, 0.0070, 0.0120 and 0.0630, respectively. Remember from Section 4.2 that each related pair contains a reference classifier trained with $t$ epochs and another classifier at epoch $t + k$. Therefore, increasing the value of k results in more changes in the parameters of the model over k epochs that are captured by our proposed method Zest.
> “

---

> > ### Comment · Reviewer_GFcZ · 2021-11-30
> > **Thanks for the responses**
> >
> > Thanks for the detailed response from the authors.

---

> ### Author Response · Authors · 2021-11-20
> **Response 2**
>
> >_**For the applications mentioned, I cannot help myself thinking of knowledge distillation, which seems to be closely related to model stealing. Is there any related work discussing this or even further, can Zest find application for knowledge distillations?**_
>
> These tasks are indeed closely related; they are both different forms of model reuse that we considered in our evaluations. We evaluated our proposed method, Zest, in detecting knowledge distillation on the ModelReuse benchmark. We also compared Zest with the related work, ModelDiff, in detecting knowledge distillation. Table 2 shows that Zest, similarly to the existing ModelDiff method, can detect knowledge distillation with $100\%$ accuracy on the ModelReuse benchmark.
>
> To better clarify the applications that we used in our experiments, we added a new Appendix A.3 that includes the description of tasks we considered. We would like to note that apart from knowledge distillation, we also evaluated Zest on transfer learning, pruning, and compression applications.
>
>
> >_**Overall I think this is a good paper and can be accepted to ICLR. Although it does not have much theoretical novelty, I believe applying LIME in the area of calculating distance between two models is pretty new. If my listed concerns can be addressed, I willing to increase my ratings one level further.**_
>
> Thanks for finding our paper new and proposing acceptance. We appreciate that you are willing to increase your rating, and hope that we have addressed your concerns.

---

### Official Review · Reviewer_vdBq · 2021-10-30

**Correctness:** 3
**Technical Novelty And Significance:** 3
**Empirical Novelty And Significance:** 3
**Recommendation:** 6
**Confidence:** 3

**Main Review:**

The paper is interesting and well written, although quite technical in scope.
All claims seem correct and quite convincing, and definitely useful (especially in the case of model stealing).
However, the validation of the proposed framework is limited to a single family of classifiers (ResNet) and a quite standard (and somewhat simple) benchmark dataset: such choice leaves room for wondering about the effectiveness of Zest on a broader landscape.
In particular, a deeper analysis of the Zest behaviour on separating related from unrelated classifiers (in different settings) would be recommended, for instance toward the definition of a (clearly empirical) quantitative distance threshold to be used as a reference for the Zest values.


**Summary Of The Paper:**

The authors propose a novel approach for computing an architecture-independent distance metric to assess the similarity between ML models by comparing the global behaviour of the models as approximated through LIME. They demonstrate the method by applying it on instances of the CIFAR dataset, focussing on the two tasks of model stealing and machine unlearning.

**Summary Of The Review:**

Paper is interesting, adding a significant novel contribution to the existing literature and providing an effective solution in at least two important and somehow still open ML issues. A wider analysis (both in terms of models and data) would be recommended to strengthen the authors' claims.

---

> ### Author Response · Authors · 2021-11-20
> **Response 1**
>
> Thank you for finding our paper interesting and well written, and highlighting that our contributions are significantly novel.  In the text below we respond to your comments.
>
>
> >_**The paper is interesting and well written, although quite technical in scope. All claims seem correct and quite convincing, and definitely useful (especially in the case of model stealing). However, the validation of the proposed framework is limited to a single family of classifiers (ResNet) and a quite standard (and somewhat simple) benchmark dataset: such choice leaves room for wondering about the effectiveness of Zest on a broader landscape.**_
>
> We would like to highlight that in our submission, we evaluated Zest on five different datasets-- CIFAR-10, CIFAR-100, ImageNet, Flower102, and SDog120-- and four classifiers from two different families-- ResNet and MobileNet. However, we agree with the reviewer that further experiments on other domains can better demonstrate Zest's effectiveness on a broader landscape. Therefore, we extended our evaluation to two new domains, text and audio, and included two new classifiers from different families, an LSTM-based model and an M5 network.
>
> We also want to point out that beyond our experiments on model stealing and verifying unlearning, our results on the ModelReuse benchmark show that Zest is applicable to detecting a wide variety of other tasks such as knowledge distillation, transfer learning, compression, and pruning with varying configurations.
>
> We added an Appendix A to clearly describe all five vision datasets, a text dataset and an audio dataset, as well as six classifiers and ModelReuse applications. All our results consistently demonstrate the effectiveness of Zest across different domains, classifiers and tasks. These additional experiments in two new domains reinforce the broad applicability of Zest.
>
>
> >_**In particular, a deeper analysis of the Zest behaviour on separating related from unrelated classifiers (in different settings) would be recommended, for instance toward the definition of a (clearly empirical) quantitative distance threshold to be used as a reference for the Zest values.**_
>
> As the reviewer expects, choosing a Zest distance threshold to separate related from unrelated classifiers is highly application specific and will likely depend on empirical knowledge. For example, a pair of models trained on the same dataset is more related than a pair of models trained on different datasets, but less so than a model and its stolen copy.
>
> However, in our experiments we found that empirically determining a threshold that differentiates related from unrelated models is generally easy. To demonstrate this, in all the figures in our manuscript, we not only plot the average Zest distances of compared models, but also include confidence intervals. In each of our experiments, across multiple architectures, datasets and tasks in the vision, text and audio domains, the confidence intervals of related and unrelated models do not overlap; thus, users can simply choose a suitable threshold between the intervals that best suits their need. We hope this demonstrates both the consistency of Zest, and that finding a working threshold in practice will likely not pose a difficult problem.
>
>
> >_**Paper is interesting, adding a significant novel contribution to the existing literature and providing an effective solution in at least two important and somehow still open ML issues. A wider analysis (both in terms of models and data) would be recommended to strengthen the authors' claims.**_
>
> We provided wider analyses and experiments in terms of both models and datasets by showing the effectiveness of Zest on the text and audio domains, as mentioned above. In addition to this, we performed an ablation study on the value of $k$ to analyse the performance of Zest on the related models as a function of $k$ in Appendix D, as follows:
>
> “
> In this section, we perform an ablation study on the value of $k$ for both CIFAR-10 and CIFAR-100. Figure 7 shows the Zest distances between related models (i.e. the same model's checkpoints at epoch $t$ and epoch $t+k$, where $k$ is set to $1, 10, 20, 50$ respectively), and between unrelated models. We can observe that
> 1. The Zest distances between related models obtained with different k values are significantly smaller than the Zest distances between unrelated models.
> 2. The larger the value of k, the larger the Zest distance between related models. For example, the Zest distance between related CIFAR-10 models for k values of 1, 10, 20 and 50 are 0.0035, 0.0070, 0.0120 and 0.0630, respectively. Remember from Section 4.2 that each related pair contains a reference classifier trained with $t$ epochs and another classifier at epoch $t + k$. Therefore, increasing the value of k results in more changes in the parameters of the model over k epochs that are captured by our proposed method Zest.
> “

---

### Official Review · Reviewer_9qy5 · 2021-11-02

**Correctness:** 3
**Technical Novelty And Significance:** 4
**Empirical Novelty And Significance:** 4
**Recommendation:** 8
**Confidence:** 3

**Main Review:**

The method is generally sensible and the experimental results look solid. The idea of using a technique which is somewhere between working in weight space and working strictly with input/output examples does make sense. Zest can distinguish between models which are snapshots of the same training process at different epochs as opposed to models trained on the same data but with initial starting points, which I think is fairly impressive, given than this distinction might be expected to be fairly subtle. Zest succeeds at detecting model extraction (aka stealing) where a competing method (ModelDiff) fails. The ability to distinguish between true unlearning and approximate unlearning is also encouraging. The paper is mostly well written, although there are some typos and some places where I would suggest edits (see below).

My main objection is that I would have liked to see Zest demonstrated on a domain other than image recognition. Testing on another domain would give me more confidence in the results, particularly given that Zest is model agnostic and should therefore be able to demonstrate success on a wide variety of domains and model classes. I also think the model stealing problem has relevance for smaller-scale tabular ML models used in social science/finance areas like credit screening, crime recidivism forecasting, etc and it would have been great to see something like that here, tested presumably on some sort of tree ensemble like gradient boosting.

I also think the applications (model similarity for model stealing, unlearning), while certainly cutting-edge, are only of moderate importance to ML research right now.

On the balance, I think the paper should probably be accepted but the limited range of experiments and moderate topic significance prevent me from giving it a very high grade.

*** Update after author rebuttal *** The authors have added additional experiments in text and audio and have offered additional information regarding the significance of the application domains ( model stealing and unlearning). For these reasons, I am increasing my score to an 8.

Some suggested edits:

Page 2:
2nd bullet point.  Also, on CIFAR-100, -< add comma
3rd bullet again need comma after CIFAR-100,
An 0% -> a 0%
Throughout our discussion, however -> throughout our discussion. However,  (start a new sentence here)
Past work on measuring model similarity have -> Past work on measuring model similarity has

Page 3
weights of a DNN captures-> weights of a DNN capture

Page 5
Are in the same size-> are the same size

Page 7
This is again over-performed by Zest -> this problem is solved by Zest.

Page 8

"To further our evaluation, we also compare our proposed method Zest using the model reuse bench mark introduced with ModelDiff (Li et al., 2021). To ensure a fair comparison, we use the Mod elReuse benchmark provided by ModelDiff:" I think this sentence is basically being repeated, unnecessarily?

Table 4 shows that Zest and ModelDiff have similar performance…I think this is Table 2, not Table 4?

**Summary Of The Paper:**

The paper presents a method (Zest) for measuring the distance or similarity between 2 supervised machine learning models while only requiring black-box access to the model's inputs and outputs but while also providing various improvements over the method of simply comparing model outputs themselves on a finite set of data points. These improvements are achieved by fitting locally linear models in the region of various reference data points, following the well-known LIME algorithm which was originally introduced for the purpose of model explanations. Experiments using DNNs for image recognition on CIFAR benchmark datasets are presented. The method is demonstrated on 2 specific applications. The first is the detection of model stealing, where the input/output examples of a model are used to reproduce it and therefore steal the associated intellectual property.  The second is the evaluation of unlearning, where a model needs to "forget" specific data and the Zest method is able to distinguish between 2 different unlearning methods in terms of their success in unlearning properly.

**Summary Of The Review:**

The Zest method presented here is sensible, clearly presented and shown to be successful for a couple of applications (model stealing and unlearning evaluation). It would have been stronger if it had been tested additionally on another problem domain and another ML model class rather than focusing solely on DNNs and image recognition. The topics are also only of moderate significance. For these reasons, I recommend a weak accept.

---

> ### Author Response · Authors · 2021-11-20
> **Response 1**
>
> We thank the reviewer for the comments, and for highlighting that our method is sensible and solid in its experiments. In the following, we respond to your comments inline.
>
>
> >_**My main objection is that I would have liked to see Zest demonstrated on a domain other than image recognition. Testing on another domain would give me more confidence in the results, particularly given that Zest is model agnostic and should therefore be able to demonstrate success on a wide variety of domains and model classes.**_
>
> We thank the reviewer for this suggestion. We performed experiments to evaluate Zest in both the text and audio domains and added a new Section 4.4 describing the setting and results as:
>
> "
> In this section, we analyse the performance of Zest in computing distances between related and unrelated model pairs in both text and audio domains. In the text domain we used Long-Short Term Memory (LSTM) classifiers (Hochreiter & Schmidhuber, 1997) and AG News dataset (Zhang et al., 2015) (described in Appendix A.1 and A.2). In the audio domain we used M5 speech classifier (Daiet al., 2017) and Speech Commands dataset (Warden, 2018). Similarly to our vision experiment in  Section  4.2, related LSTM (or  M5) classifiers are checkpoints at two different epochs, while unrelated LSTM (or M5) classifiers are trained with different random seeds. Figure 4 shows that the distance output by Zest for unrelated classifiers in both text and audio domains is significantly higher than for related classifiers (consistent with our results in Section 4.2 on vision datasets). This thus reinforces that Zest is applicable to different domains of vision, audio and text.
> "
>
> In summary as detailed in Appendix A, we evaluated Zest in seven different datasets (five of these are vision datasets, one a text dataset and the last an audio dataset) as well as six different classifiers (family of ResNet, MobileNet, LSTM and M5).
>
> These evaluations demonstrate the success of Zest on a wider variety of domains and model architectures.
>
>
> >_**I also think the model stealing problem has relevance for smaller-scale tabular ML models used in social science/finance areas like credit screening, crime recidivism forecasting, etc and it would have been great to see something like that here, tested presumably on some sort of tree ensemble like gradient boosting.**_
>
> We agree that the model stealing problem has relevance for these domains. Due to the time limit, we chose to prioritize further evaluating our method on LSTM classifiers in the text domain and M5 speech classifiers in the audio domain, and leave the evaluation on tabular data and gradient boosting classifiers as future work.
>
> >_**I also think the applications (model similarity for model stealing, unlearning), while certainly cutting-edge, are only of moderate importance to ML research right now.**_
>
> In addition to model stealing and unlearning, our experiments on the ModelReuse benchmark suggest that Zest can detect 14 other reuse tasks, such as transfer learning, pruning, compression and knowledge distillation (please see the results in Table 2 and the description in Appendix A.3). Below we highlight the importance of model staling and unlearning.
>
> **Model stealing**: Model stealing is one of the most important threats that companies worry about (see Table V in [1] and the statement of retail organization “We run a proprietary algorithm to solve our problem and it would be worrisome if someone can reverse engineer it” in Section 2 of [1]). Trained models are usually confidential given the difficulty of collecting training data as well as training the model. Therefore, stealing the model can lead to the loss of intellectual property. Furthermore, model stealing can be used as a reconnaissance step prior to mounting further attacks (e.g., finding adversarial examples that transfer back to the stolen model).
>
> **Unlearning**: Apart from the security and privacy community [2], there have been growing interest in unlearning in the ML community such as ICML [3] and NeurIPS [4,5,6]. We also would like to point out that we are the first to verify unlearning using model similarity, while previous works mostly focus on using model similarity to detect model stealing or reuse.
>
>
>
> [1] Shankar Siva Kumar et al. Adversarial Machine Learning - Industry Perspectives, S&P Workshops, 2021.
>
> [2] Bourtoule et al. Machine Unlearning, S&P 2021.
>
> [3] Brophy and Lowd. Machine Unlearning for Random Forests, ICML 2021.
>
> [4] Acharya et al. Remember What You Want to Forget: Algorithms for Machine Unlearning, NeurIPS 2021.
>
> [5] Gupta et al. Adaptive Machine Unlearning, NeurIPS 2021.
>
> [6] Ginart et al. Making AI Forget You: Data Deletion in Machine Learning, NeurIPS 2019
> .

---

> ### Author Response · Authors · 2021-11-20
> **Response 2**
>
> >_**On the balance, I think the paper should probably be accepted but the limited range of experiments and moderate topic significance prevent me from giving it a very high grade.**_
>
> Thank you, to summarize our changes we added most importantly:
> 1. new analyses of our proposed method Zest in Appendix D;
> 2. an evaluation of Zest in other domains in Section 4.4.
>
> Regarding the new analyses, we added a new Appendix D in the manuscript to show the results of our ablation study on other values of k (k = 1, 10, 20, 50) as follows:
>
> “
> In this section, we perform an ablation study on the value of $k$ for both CIFAR-10 and CIFAR-100. Figure 7 shows the Zest distances between related models (i.e. the same model's checkpoints at epoch $t$ and epoch $t+k$, where $k$ is set to $1, 10, 20, 50$ respectively), and between unrelated models. We can observe that
> 1. The Zest distances between related models obtained with different k values are significantly smaller than the Zest distances between unrelated models.
> 2. The larger the value of k, the larger the Zest distance between related models. For example, the Zest distance between related CIFAR-10 models for k values of 1, 10, 20 and 50 are 0.0035, 0.0070, 0.0120 and 0.0630, respectively. Remember from Section 4.2 that each related pair contains a reference classifier trained with $t$ epochs and another classifier at epoch $t + k$. Therefore, increasing the value of k results in more changes in the parameters of the model over k epochs that are captured by our proposed method Zest.  “
>
> Regarding new experiments, we evaluated Zest in both the text and audio domains and added a new Section 4.4 describing the setting and results as (this is the same paragraph referred in Response 1):
>
> “In this section, we analyse the performance of Zest in computing distances between related and unrelated model pairs in both text and audio domains. In the text domain, we used Long-Short term memory (LSTM) classifiers (Hochreiter & Schmidhuber, 1997) and AG News dataset (Zhang et al.,2015) (described in Appendix A.1 and A.2). In the audio domain, we used M5 speech classifier (Daiet al., 2017) and Speech Commands dataset (Warden, 2018).  Similarly to our vision experiment in  Section  4.2,  related  LSTM  (or  M5)  classifiers are checkpoints at two different epochs,  while unrelated LSTM (or M5) classifiers are trained with different random seeds.  Figure 4 shows that the distance output byZestfor unrelated classifiers in both text and audio domains is significantly higher than for related classifiers (consistent with our results in Section 4.2 on vision datasets). This thus reinforces thatZestis applicable to different domains of vision, audio, and text.’’
>
> In summary, as detailed in Appendix A, we evaluated Zest in seven different datasets (five of these are vision datasets, one a text dataset and the last an audio dataset) as well as six different classifiers (family of ResNet, MobileNet, LSTM and M5).
>
>
>
> >_**Some suggested edits**_
>
> Thank you for your suggestions, we implemented them all.

---

### Official Review · Reviewer_MKjr · 2021-11-04

**Correctness:** 3
**Technical Novelty And Significance:** 1
**Empirical Novelty And Significance:** 2
**Recommendation:** 3
**Confidence:** 4

**Main Review:**

Strengths.

The general method of the paper is presented relatively clear, and a series of experiments have shown the advantage of the presented method.

Weaknesses.

Based on the introduction, it seems that existing literature on measuring model distances only includes Li et al. 2021 and Jia et al. 2021. The authors directly point out the limit of these two papers and propose a new method. It is hard to evaluate the challenges of this research area without more context.

The paper mentions that Zest can also be applied to other domains like text and audio. Given that this paper focuses more on the empirical side, it would be good to include more diverse experiments and add applications in text/audio domains into the experiment section of the paper.

Figure 1 is not clear enough on its own, especially Phase 3 & 4, and the caption does not contain enough information/description to help the audience understand directly. Also, it would be better if the LIME part can be more abstract since LIME algorithm is not this paper's contribution (similarly for Algorithm 1).

Regarding class alignment step in Algorithm 1, does any of the experiments use class alignment? In general, when the two classifiers have unrelated labels, how could this alignment step work? More explanations on this may be required.

**Summary Of The Paper:**

The paper proposes an architecture-independent way of measuring distance between two models for the same type of input data. More specifically, given each reference sample, it first calculates a linear weight based on a set of masked samples and their corresponding outputs for each model, then calculates the distance between the set of weights for all reference samples generated from the two models. The method can be applied to help detect model stealing and inform unlearning verification.

**Summary Of The Review:**

The paper is an empirical study on a topic that is not very popular (at least based on how the paper discusses about the existing literature).

---

> ### Author Response · Authors · 2021-11-20
> **Response 1**
>
> We thank the reviewer for the feedback. In the text below we respond to the comments which we have grouped by topic.
>
> >_**Based on the introduction, it seems that existing literature on measuring model distances only includes Li et al. 2021 and Jia et al. 2021. The authors directly point out the limit of these two papers and propose a new method. It is hard to evaluate the challenges of this research area without more context.**_
>
> The reviewer is right that, as far as we are aware, Li et al. 2021 and Jia et al. 2021 are the only published works directly related to this topic. However, we believe this new research area is important due to several immediate applications. Beyond model stealing and unlearning, which we predominantly focused on in our work, measuring model distances is also useful for detecting other methods of model reuse (e.g. transfer learning, distillation, pruning, etc.) to protect intellectual property, and for the prevention of vulnerability propagation during transfer learning. The latter issue is identified by [1], who demonstrated that vulnerabilities like backdoors are propagated from pre-trained models to fine-tuned models. Our results on the ModelReuse benchmark show that Zest outperforms these prior approaches in these domains as well. Finally, we would like to refer to a comment from fellow reviewer vDbq who confirms the importance of this topic and our solution: “the paper is interesting, adding a significant novel contribution to the existing literature and providing an effective solution in at least two important and somehow still open ML issues.”
>
> Following your remark, we added more context about the challenges of this topic in the first paragraph of the introduction by changing
> “Comparing the functional behavior of ML models is often challenging.”
>
> to
>
> “Comparing the functional behavior of ML models is often challenging because they are not easily inspectable (e.g. located in the cloud). In addition, ML models that capture similar knowledge may not share similar architectures, and vice versa, making it difficult to compare models in weight space. Finally, ML models that solve different tasks may nonetheless share similar knowledge, such as in the case of transfer learning (Li et al. 2021)."
>
> [1] Davchev, T., Korres, T., Fotiadis, S., Antonopoulos, N., & Ramamoorthy, S. An empirical evaluation of adversarial robustness under transfer learning. ICML workshop, 2019.
>
> >_**The paper mentions that Zest can also be applied to other domains like text and audio. Given that this paper focuses more on the empirical side, it would be good to include more diverse experiments and add applications in text/audio domains into the experiment section of the paper.**_
>
> We thank the reviewer for this suggestion. We performed experiments to evaluate Zest in both text and audio domain and added a new Section 4.4 describing the setting and results as:
>
> "In this section, we analyse the performance of Zest in computing distances between related and unrelated model pairs in both text and audio domains. In the text domain, we used Long-Short term memory (LSTM) classifiers (Hochreiter & Schmidhuber, 1997) and AG News dataset (Zhang et al.,2015) (described in Appendix A.1 and A.2). In the audio domain, we used M5 speech classifier (Daiet al., 2017) and Speech Commands dataset (Warden, 2018).  Similarly to our vision experiment in  Section  4.2,  related  LSTM  (or  M5)  classifiers are checkpoints at two different epochs,  while unrelated LSTM (or M5) classifiers are trained with different random seeds.  Figure 4 shows that the distance output byZestfor unrelated classifiers in both text and audio domains is significantly higher than for related classifiers (consistent with our results in Section 4.2 on vision datasets). This thus reinforces thatZestis applicable to different domains of vision, audio, and text.’’
>
> In summary, as detailed in Appendix A, we extended the evaluation of Zest to seven different datasets (five of these are vision datasets, a text dataset, and an audio dataset) and six different classifiers (family of ResNet, MobileNet, LSTM, and M5) in total.

---

> ### Author Response · Authors · 2021-11-20
> **Response 2**
>
> >_**Figure 1 is not clear enough, the caption does not contain enough information/description, abstract LIME in algorithm and Figure1.**_
>
> Thank you for pointing this out. We expanded the caption of Figure 1 to improve its clarity as
>
> “
> An overview of the Zest approach to computing the distance between two black-box classifiers $C_1$ and $C_2$. Phase 1 randomly samples several reference images. Phase 2 segments the reference images into components and their location masks using a segmentation model. Phase 3 approximates the local decision boundaries around each reference image by training a LIME model that takes mask locations as input and predicts the classifier's response on these images’ components. Finally, Phase 4 combines the local decision boundaries of all reference points to approximate the global decision boundaries of $C_1$ and $C_2$, and computes their Cosine distance.
> “
>
> We also agree with the reviewer that LIME is not our contribution, and have considered abstracting its details from Figure 1 and Algorithm 1 to avoid misleading readers, and for simplification. However, we ultimately chose to include them because we think the description of its inner functionality provides essential intuition on why our method works. We hope that our explanations in the introduction and background sections make it sufficiently clear that we do not claim any credit for LIME, but instead, build on it to develop a new approach to compute model distances.

---

> ### Author Response · Authors · 2021-11-20
> **Response 3**
>
> >_**Regarding the class alignment step in Algorithm 1, does any of the experiments use class alignment? In general, when the two classifiers have unrelated labels, how could this alignment step work? More explanations on this may be required.**_
>
> We use class alignment in all experiments. We added an Appendix B  description of how we perform class alignment in each setting:
>
> “We evaluate our class alignment strategy in two different settings of comparing two classifiers $C_1$ and $C_2$:
> 1. Both models are trained with the same architecture and on the same dataset but with different output label orders;
> 2. Each model is trained with the same architecture on a different dataset and with a different number of labels;
>
> In setting (1), $C_1$: is trained on CIFAR10 (or CIFAR100) and $C_2$: trained on CIFAR10 (CIFAR100) but with randomly swapped labels. Our experimental results show that our class alignment strategy is always able to correctly match the classes of two models in this setting. (e.g. matching the “cat” classes of the two models when the “cat” class of one of the models is indexed by 3 while the “cat” class in the other model is indexed by 7). The intuition behind this is that class alignment finds the class pairs with the most similar LIME masks, in other words, the class predictions are made using similar features of the inputs.
>
> Setting (2) includes the transfer learning task of ModelReuse experiments. In this setting, models are trained on two different datasets with a different number of labels (1000 classes for ImageNet versus 120 for SDog120, or 1000 classes for ImageNet versus 102 for Flower102). These two datasets however share some common classes such as breeds of dog or type of flowers. For example in one of the experiments, $C_1$ is pre-trained on ImageNet with 1000 classes and $C_2$ is fine-tuned $C_1$ on Flower102 with 102 classes. The class alignment of these two classifiers with unrelated labels are as follows:
> 1. Train LIME linear models of $C_1$ using 128 reference data points, and train LIME linear models of $C_2$ using 128 reference data points.
> 2. Concatenate weights of all LIME linear models of $C_1$ class-wise to obtain $S_1={W_1^{p_1}|...|W_1^{p_128}}$ as the order of labels are fixed, and concatenate weights of all LIME linear models of $C_2$ class-wise to obtain $S_2={W_2^{p_1}|...|W_2^{p_128}}$.
> 3. Setting a reference classifier: we choose the classifier $C_2$ that includes the lower number of classes (102 labels) as a reference classifier;
> 4. Compute the Cosine distance between each row of $S_2$ with all rows of $S_1$. We do this because we assume that the semantic of labels and number of common labels are unknown.
> 5. Pick 102 pair rows with the minimum distances, determining 102 classes chosen out of 1000 classes of $C_1$. This helps us to compare $C_1$ and $C_2$ based on the classes that perform most similarly to each other.
>
> This means that we might have pairs of aligned classes that are not related semantically, but this is expected and will contribute to telling the classifiers apart distance-wise.
>
> In another example of the ModelReuse transfer learning experiments, we compare a ResNet18 trained on SDog120 with another ResNet18 trained on ImageNet. As mentioned in Appendix A, SDog120 contains 120 breeds of dogs. On the other hand, ImageNet includes 130 breeds of dogs out of its 1000 classes. In this experiment, our class alignment is able to pick 120 similar breeds of dog from ImageNet which are most similar to the 120 breeds of dogs from SDog120.”

---

> ### Author Response · Authors · 2021-11-24
> **Follow up on our responses**
>
> Dear Reviewer MKjr,
>
> Thank you for your time reading our responses. We were wondering if we were able to adequately address your concerns.
>
>
> Best,
>
> Paper3112 authors

---

### Decision · Program_Chairs · 2022-01-20

**Decision:**

Accept (Poster)

**Comment:**

This paper presents a method, called Zest, to measure the similarity between two supervised machine learning models based on their model explanations computed by the LIME feature attribution method.  The technical novelty and significant are high, and results are strong.  Reviewers had clarifying questions regarding experiments and suggestions to add experiments, which involve additional domains (text and audio) and different families of classifiers, and more contexts based on prior literatures. These were adequately addressed by the authors. Overall, this paper deserves borderline acceptance.